# Metabolic Comorbidities in Vitiligo: A Brief Review and Report of New Data from a Single-Center Experience

**DOI:** 10.3390/ijms22168820

**Published:** 2021-08-17

**Authors:** Andrea D’Arino, Mauro Picardo, Mauro Truglio, Alessia Pacifico, Paolo Iacovelli

**Affiliations:** 1Laboratory of Cutaneous Physiopathology and Integrated Center of Metabolomics Research, San Gallicano Dermatological Institute, IRCCS, 00144 Rome, Italy; andrea.darino@ifo.gov.it (A.D.); mauro.truglio@ifo.gov.it (M.T.); 2Clinical Dermatology, Phototherapy Unit, San Gallicano Dermatological Institute, IRCCS, 00144 Rome, Italy; alessia.pacifico@ifo.gov.it (A.P.); paolo.iacovelli@ifo.gov.it (P.I.)

**Keywords:** vitiligo, metabolic syndrome, atherosclerosis, cell–cell cross-talk, melanocytes, adipocytes, comorbidities

## Abstract

Among disorders of pigmentation, vitiligo is the most common, with an estimated prevalence between 0.5% and 1%. The disease has gathered increased attention in the most recent years, leading to a better understanding of the disease’s pathophysiology and its implications and to the development of newer therapeutic strategies. A better, more integrated approach is already in use for other chronic inflammatory dermatological diseases such as psoriasis, for which metabolic comorbidities are well-established and part of the routine clinical evaluation. The pathogenesis of these might be linked to cytokines which also play a role in vitiligo pathogenesis, such as IL-1, IL-6, TNF-α, and possibly IL-17. Following the reports of intrinsic metabolic alterations reported by our group, in this brief review, we analyze the available data on metabolic comorbidities in vitiligo, accompanied by our single-center experience. Increased awareness of the metabolic aspects of vitiligo is crucial to improving patient care.

## 1. Introduction

Among disorders of pigmentation and specifically diseases characterized by true depigmentation, vitiligo is certainly the most common, with the worldwide prevalence estimated around 0.5% and 1%. Clinically, it presents with well-demarcated milky-white macules characterized by variable dimensions and different distribution patterns. Its existence was recognized very long ago, and initial descriptions date back to the second millennium BCE [1]. However, notwithstanding its presence for several millennia and being characterized by cell death, vitiligo has been a rather neglected disease in dermatology. Still, today, doctors consider it a simple aesthetic problem, and treatment turns out to be non-satisfactory. The burden of depigmentation for patients has been well-demonstrated, with several reports linking vitiligo with anxiety or depression [2,3]. However, in the past few years, this scenario seems to have finally changed. In 2012, vitiligo was finally classified by the Vitiligo European Taskforce (VETF) to simplify the nomenclature [4]. Furthermore, giant steps have been made in the understanding of the pathophysiology of vitiligo with particular progress in the elucidation of immunologic mechanisms [5]. 

Several pathogenetic hypotheses have been elaborated throughout the years. The former theory describes pigment loss as a direct consequence of aberrant melanin synthesis through the formation of toxic intermediates [6]. Another theory, known as the neural theory, laid its basis on the neural crest origin of melanocytes and the presence of neurogenic inflammatory mediators which have shown melanocyte toxicity such as NGF and NPY [7]. Still, notwithstanding the historical importance of these theories, the increasing attention gained by vitiligo and the developing medical research quickly led to their abandonment. The immune hypothesis stemmed from the frequent observations of T cell activity in vitiligo progression which is supported by genetic data [8]. Recently, the biochemical hypothesis has been proposed, suggesting that intracellular oxidative stress could be the initial event for melanocyte destruction [9].

The convergence theory was subsequently elaborated to unite all the existing theories in a more comprehensive, wide pathogenetic view of the disease (Figure 1) [10]. This theory is supported by the recent research and involves genetic susceptibility and the presence of intrinsic abnormalities in melanocytes. These are responsible for greater susceptibility to oxidative stress damage which leads to the release of factors that induce immune activation and, finally, melanocyte loss [5]. Our group has been particularly active in the elucidation of nonimmunological factors in the pathogenesis of vitiligo, confirming the presence of metabolic abnormalities in almost all epidermal cells. Starting from melanocytes, the obvious suspects in vitiligo, we have consistently shown the presence of a disruption of the redox balance characterized by excessive production of radical oxygen species (ROS) and inefficiency in antioxidant mechanisms [11]. Furthermore, vitiligo melanocytes in vitro are characterized by impaired energetic metabolism which is defined by an ineffective effort in compensating defective ATP production through increased glucose utilization [12]. Membrane lipid alterations such as reduced cardiolipin and increased cholesterol have also been found in the inner mitochondrial membrane [13]. Increased ROS production, reduced antioxidant capacity, mitochondrial dysfunctions, and caspase 3 activation have also been shown in perilesional keratinocytes [14]. Finally, increased ROS levels have also been shown in vitiligo fibroblasts, which are responsible for the induction of a myofibroblast-like phenotype. All these abnormalities have been linked to the activation of the immune response. Today, it is well-accepted that the IFNγ signaling pathway is the key player of the disease’s pathogenesis. Tulic et al. demonstrated that both natural killer (NK) and innate lymphoid cells (ILCs) can produce increased amounts of IFNγ in the presence of ROS, Hsp70i, and HMGB1 [15]. The production of IFNγ subsequently induces the release of key chemokines CXCL9 and CXCL10, particularly from the neighboring keratinocytes which are responsible for Th1/Tc1 cell recruitment, especially in perilesional skin [16,17]. More recently, a subpopulation of T cells has drawn increasing attention: T resident memory cells (T_RM_). These cells, which are characterized by the co-expression of CD69, CD103, and CD49a [18], can persist in tissues for rather long periods after the appearance of the disease and rapidly reactivate an immune response upon stimulation. Their presence has been linked to the frequent recurrence of vitiligo lesions after successful therapy in the same areas which have been treated and could play a role in the occurrence of the Koebner phenomenon. The presence of a systemic inflammatory process in vitiligo is confirmed by several observations. In consideration of the IFNγ signature, chemokines CXCL9 and CXCL10 have been investigated as possible circulating biomarkers of disease activity. Both chemokines have been associated with progressive cases, with a stronger association demonstrated for CXCL10 [19]. In non-lesional skin of active vitiligo patients, CXCL10 has been found at higher levels [20], as well as both CXCL9 and CXCL10 mRNA by immunohistochemistry [21]. Several cytokines have also been reported to be increased in vitiligo patients, and IL-17 has been consistently reported at higher levels, as recently reviewed by Singh et al. [22]. On the contrary, the data on IL-6 are more conflicting, while some link with disease activity has been shown for TNF-α [23].

The greater knowledge and the increased attention to vitiligo lead to a more complete view of all the disease’s implications, including the possibility of other concomitant diseases with a possible common pathogenetic pathway.

Several reports regarding the coexistence of other diseases in patients with vitiligo are present in the literature, with more consistent ones existing for autoimmune comorbidities [24]. Overall, thyroid disease, specifically autoimmune thyroid disease, is recognized as the most established association with prevalence rates which vary between 1% and 37% in adult patients [25,26] and up to 6% in children [27] according to different studies. Other reported autoimmune diseases include alopecia areata, diabetes mellitus, pernicious anemia, systemic lupus erythematosus, rheumatoid arthritis, Addison’s disease, and inflammatory bowel disease, albeit with less significant prevalence rates.

A more integrated approach is already in use for other inflammatory dermatological conditions. This has led to the demonstration of increased comorbidities in patients affected by psoriasis [28], particularly of cardiovascular diseases, metabolic syndrome, and increased atherosclerosis [29]. Similar findings have been reported for lichen planus [30,31,32,33], lupus erythematosus [34], Sjögren syndrome [35], chronic urticaria [36], and vasculitis. For psoriasis, it is interesting to note that it probably acts as an independent cardiovascular risk factor and, as such, the increased association with cardiovascular diseases is directly correlated with the severity of the disease, and it is not only dependent on the association of psoriasis with some or even all components of the metabolic syndrome [37]. Realistically, chronic skin inflammation alone is not enough to justify these observations as suggested by studies for atopic dermatitis which failed to demonstrate a similar association [38]. Thus, systemic inflammation, such as high levels of IL-17 in patients with acute coronary syndrome and psoriasis, is probably involved [39]. Furthermore, increased body fat and obesity are associated with elevated serum free fatty acids (FFA), which increase the sensitivity of dendritic cells to the amplification of Th1/Th17 responses [40], and an increase in circulating IL-17 [41]. The role of IL-17 has also been suggested in vitiligo: in humans, increased levels of circulating Th17 cells have been reported [42]. These are a subset of CD4+ T cells which secrete several cytokines, including IL-17, which has been shown at higher levels in vitiligo patients, correlating with disease duration, extent, and activity [22]. It would be possible to hypothesize parallelism between psoriasis and vitiligo in which high IL-17 levels could contribute to the presence of cardiovascular comorbidities in the latter group of patients.

Another disease with a similar pathogenetic fingerprint is alopecia areata, in which IFNγ drives the immune response in the context of ectopic expression of MHC class I molecules which exposes the hair follicle autoantigen to autoreactive T cells [43]. Additionally, Th17 cytokines, such as IL-17A, have also been associated with systemic autoimmune activation in the disease [44]. The coexistence of obesity and metabolic syndrome has also been reported for alopecia areata [44], and recently adiponectin has been proposed as a marker of disease activity [45].

Increased awareness of the metabolic aspects of vitiligo is crucial to improve patient care. In this brief review, we analyze the available data on metabolic comorbidities in vitiligo, accompanied by our single-center experience.

## 2. Metabolic Comorbidities in Vitiligo

Known in the past under a variety of other names such as syndrome X, insulin resistance syndrome, or the obesity dyslipidemia syndrome [46], the term metabolic syndrome is today defined as the cooccurrence of metabolic risk factors for cardiovascular disease and type 2 diabetes [47]. The former is specifically characterized by abdominal obesity, hyperglycemia, dyslipidemia, and hypertension. The most widely adopted definition was given by the National Cholesterol Education Program (NCEP) Adult Treatment Panel III (ATP III) [48]. The prevalence differs between countries depending on the diagnostic criteria used and geographic factors, reaching peaks of up to 50% of the over-60 population in the United States [49]. The presence of proinflammatory cytokines, such as IL-6, IL-1, and TNF-α, has been linked to the pathogenesis of insulin resistance and endothelial dysfunction in metabolic syndrome, contributing to the presence of diffuse subclinical inflammation [50,51]. 

Interestingly, IL-6, IL-1, and TNF-α, which have been associated with insulin resistance and atherosclerosis, are also among the cytokines which play a role in the pathogenesis of vitiligo, possibly highlighting a link between these two conditions (Figure 2). 

These cytokines are also known as adipocytokines in consideration of their adipocyte origin [52]. Adipose tissue is recognized today as a true endocrine and paracrine organ which plays the key role in the induction of the syndrome. IL-6 is responsible for insulin resistance [53] and has been associated with hypertension and atherosclerosis in murine models [54] while TNF-α is a mediator of atherosclerosis and heart failure [55]. Interestingly, the secretion of these cytokines has been correlated to obesity-induced ROS production, as shown by some in vitro studies [56], possibly highlighting one more link between the pathogenesis of the metabolic syndrome and vitiligo. Moreover, the anti-inflammatory cytokine adiponectin, normally produced by adipocytes, is decreased in patients affected by the syndrome leading to reduced insulin sensitization, anti-atherogenesis, and vasodilation [57].

Studies in the literature investigating this association are rare, and most of them have been published in recent years as there is growing attention to the pathogenesis of vitiligo and better recognition of the disease from international communities. A Turkish prospective cross-sectional study carried out by Tancan et al. between 2014 and 2016 on 155 vitiligo patients and controls showed a statistically significantly higher incidence of the metabolic syndrome classified according to the ATP III criteria in patients with vitiligo, especially those with a higher Vitiligo Area Severity Index (VASI) [58]. In the vitiligo patients, fasting glucose, LDL cholesterol, and blood pressure levels were all significantly higher than in the controls even after the exclusion of the patients affected by comorbidities such as obesity, diabetes, and Hashimoto’s thyroiditis. Interestingly, the mean fasting glucose was 93 mg/dL, a level which does not classify for overt diabetes nor impaired fasting glucose but can be considered within the higher spectrum of the norm. Overall, criteria for metabolic syndrome were met in 37.4% of vitiligo patients. Similar results were reported in another Turkish case–control study which showed an increased risk of developing the metabolic syndrome in patients with vitiligo and correlations between the syndrome and the activity and duration of vitiligo [59]. Another study group from Iran showed that vitiligo patients had higher levels of fasting blood glucose, total cholesterol, waist circumference, and LDL while HDL levels were generally lower, and the syndrome was diagnosed in more vitiligo patients than in the healthy controls. Moreover, the researchers performed high-resolution ultrasonography to assess the intima–media thickness of the common carotid artery as a marker of subclinical atherosclerosis. Overall, vitiligo patients showed higher frequencies of subclinical atherosclerosis than healthy subjects [60]. 

Not all studies are concordant on the possible correlation between vitiligo disease activity and metabolic syndrome severity. Sharma et al. have successfully confirmed the increased rates of metabolic syndrome, hypertriglyceridemia, low HDL levels, and impaired glucose tolerance in patients with vitiligo but failed in demonstrating any correlation with disease severity [61]. However, this study used the VIDA score to estimate disease severity rather than the more largely adopted VASI. Karadag et al. are concordant with the above-cited papers, showing lower HDL concentration, increased LDL/HDL ratio, and higher insulin resistance as measured with homeostatic model assessment-IR (HOMA-IR) [62]. A recent study by Azzazi et al. has shown that a subset of patients with vitiligo is at a higher risk of developing dyslipidemia and atherosclerosis, which might increase their future risk for the development of cardiovascular disease, by means of analysis of the serum lipid profile, oxidative stress biomarkers, and carotid duplex [63]. The possible alterations of the lipid profile seem not to be limited to adults as Pietrzak et al. have shown increased LDL, decreased HDL, and increased LDL/HDL ratio and triglycerides in vitiligo-affected children [64]. This latter finding is of particular interest as it underscores the possibility that inflammatory factors could be behind these metabolic derangements.

While it is true that these observations could also be justified by the higher rates of metabolic syndrome elements in vitiligo patients, in most studies, these remain significantly higher than in the controls even when excluding subjects with comorbidities which could contribute to the diagnosis of the syndrome. Thus, it could be hypothesized that the systemic proinflammatory status contributes to atherosclerosis in vitiligo patients and, as mentioned above, the crucial role might be played by the key cytokines in vitiligo pathogenesis, such as IL-1, IL-6, TNF-α, and possibly IL-17.

IL-1, IL-6, and TNF-α are expressed at higher levels by keratinocytes and negatively influence melanogenesis as shown by mRNA expression studies [65]. IL-6 has also been shown to work reliably as a marker of disease activity together with CXCL10 [66]. The role of IL-17 is more controversial: while it is true that increased levels of circulating Th17 cells have been reported [42], other studies have shown no difference in IL-17 secretion between the vitiligo T cells and the healthy controls and lower IL-17 levels than in psoriasis [67]. The relative importance of IL-17 has also been confirmed in a trial evaluating secukinumab, an anti-IL-17 antibody, for the treatment of vitiligo which failed to demonstrate any significant efficacy [68].

Nevertheless, not all studies have consistently confirmed the presence of all metabolic risk factors in patients with vitiligo. Sinha et al. have recently published results that confirmed the presence of significant differences in HDL levels and triglyceride levels while waist circumference, plasma glucose, and blood pressure were comparable between the cases and the controls [69]. However, the mean fasting glucose in the vitiligo patients was 101.56 mg/dL, a value well within the range of impaired fasting glucose.

At the extreme opposite are the results by Rodrìguez-Martin et al. who have found a more favorable lipid profile in vitiligo patients, with higher HDL levels and lower triglycerides [70].

To further analyze this possible correlation, we carried out a retrospective analysis on a database containing data of the patients referred to the vitiligo unit of the San Gallicano Dermatological Institute in Rome, Italy, between January 2017 and January 2021. The ethical committee approved the study. To be included in the study, each patient needed to have a confirmed clinical diagnosis of vitiligo obtained with the aid of Wood’s lamp and categorized according to the Vitiligo Global Issues Consensus Conference (VGICC) guidelines for vitiligo classification [4]. Patients aged ≥ 18 years for whom central blood test data were available were enrolled. A subset of patients was analyzed with the aid of artificial intelligence software which was able to extract data from available PDFs when the central data were not available. The following parameters were analyzed: total cholesterol, HDL, LDL, triglycerides, and fasting blood glucose (FPG). The age and sex-matched control group comprised hospital workers undergoing routine medical fitness evaluations. A total of 839 vitiligo patients and 316 healthy controls were enrolled.

The demographic characteristics and main blood parameters are shown in Table 1. In consideration of the nonnormal distribution of the hematologic parameters, the nonparametric Mann–Whitney *U* test was carried out to analyze the parameters. When compared to the control group, the vitiligo patients consistently showed higher FPG (91.9 ± 18.9 vs. 87.9 ± 6.9), total cholesterol (299.1 ± 37.6 vs. 193.3 ± 35.6), LDL cholesterol (116.5 ± 34.1 vs. 103.7 ± 17.1), and triglycerides (121 ± 167.7 vs. 83.5 ± 26.9), while HDL cholesterol was lower (60.3 ± 19.7 vs. 64.6 ± 13.8); all the differences were statistically significant (*p* < 0.05). 

Our findings are concordant with the majority of these observations. In our sample, while the vitiligo patients had lower levels of total cholesterol overall, the distribution of lipid subpopulations was less favorable than in the healthy controls with LDL cholesterol and triglycerides showing higher values than in the healthy controls, as well as lower HDL levels. Moreover, FPG levels were also significantly higher. While these were not in the range of either overt diabetes or impaired fasting glucose, the median FPG level was 90 mg/dL, a value that can be considered within the higher end of the normal spectrum. This observation is crucial as it underlines that vitiligo patients might display slight metabolic derangements which could subsequently evolve in a more clinically relevant disease. Such results warrant a routine investigation of these parameters in daily practice as the restoration of a physiological metabolic profile could lead to better therapeutic outcomes.

As already hypothesized by Pietrzak et al., if these findings continue to be consistently reported, they could change the way we approach vitiligo treatment [71]. A more complete approach is already in use for psoriasis where the primary prevention of cardiovascular diseases has also been investigated through the evaluation of the impact of systemic drugs on cardiovascular outcomes [29]. For example, studies have shown a possible cardioprotective effect of TNF inhibition [72]. If this were to be true for vitiligo, the development of more effective systemic anti-inflammatory treatments such as JAK inhibitors could prove to be a more effective strategy in approaching the treatment of the disease. Going beyond the metabolic syndrome, a new area for further studies could be the possible association between atherosclerosis and vitiligo which has been only hinted at in the literature and could be justified by the presence of systemic inflammation [73]. 

## 3. Conclusions

In conclusion, the increased attention gained by vitiligo in recent years is slowly enabling the development of a new view of the disease that looks beyond the “aesthetic” cutaneous manifestation and the psychosocial impact. A better understanding of the pathogenesis has led to the discovery of possible new associations, particularly following reports of intrinsic metabolic abnormalities in epidermal cells of vitiligo patients. Overall, the data in the literature seem to highlight a possible association between vitiligo and alterations of the spectrum of the so-called metabolic syndrome for the most part. Our findings largely confirm what most of the published literature has already reported. These observations could become particularly relevant, especially considering the rapid development of new therapies for vitiligo, such as topical and systemic Janus kinase (JAK) inhibitors which have shown the most promising results [74,75]. Given their broad anti-inflammatory activity, a similarly wide range of potential adverse effects has been reported still notwithstanding the favorable risk–benefit ratio [76]. Moreover, other treatment targets are being explored, and several drugs are under clinical evaluation. In this scenario, vitiligo comorbidities must be evaluated in a new light taking into consideration the possible interference between these and the proposed novel treatments. Nevertheless, not all research is concordant on this aspect and further studies are probably necessary to shed light on this possible association and improve the quality of life of vitiligo patients.

## Figures and Tables

**Figure 1 ijms-22-08820-f001:**
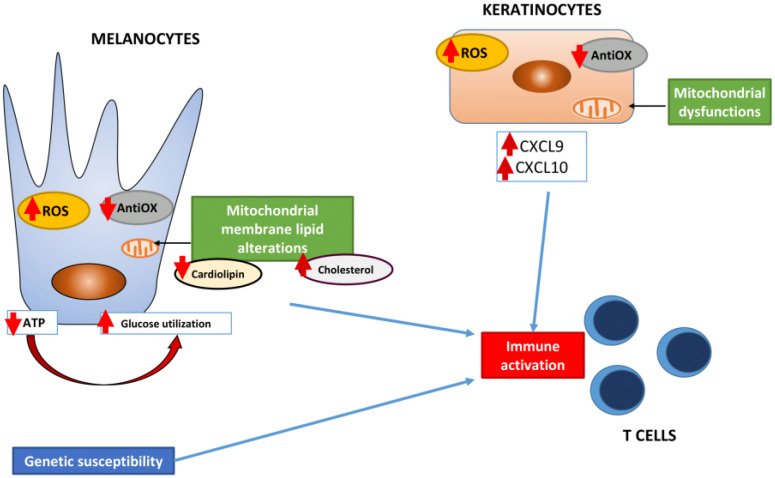
Schematic overview of the convergence theory for vitiligo pathogenesis.

**Figure 2 ijms-22-08820-f002:**
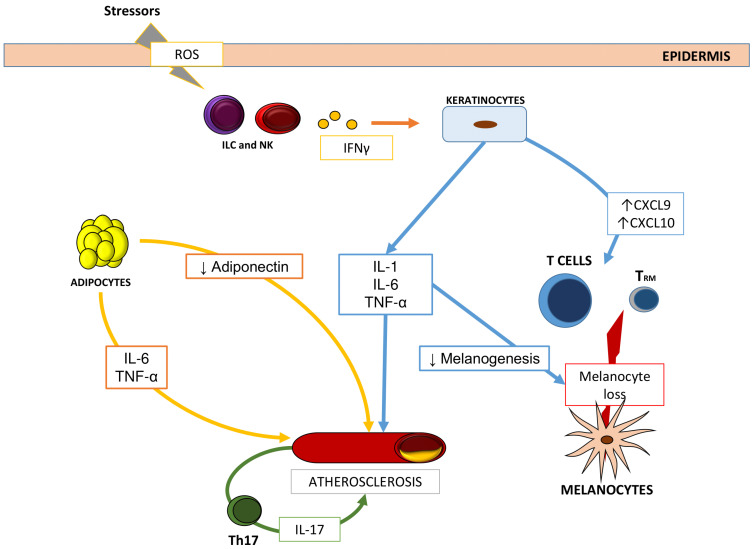
Proposed cross-relationship between vitiligo pathogenesis and metabolic syndrome manifestations, specifically atherosclerosis. Cytokines that play a role in vitiligo, such as IL-1, IL-6, and TNF-α, have been shown to induce accelerated atherosclerosis.

**Table 1 ijms-22-08820-t001:** Demographic features and serum metabolic parameters of the study and control group patients.

Variables	Vitiligo Group	Control Group	*p*
**Gender (*n*, %)**			0.42
**Female**	488 (58.2%)	192 (60.9%)	
**Male**	351 (41.8%)	124 (39.1%)	
**Age (mean ± SD, range)**	45.3 ± 15.5 (18–88)	46.8 (27–67)	
**FPG (mg/dL) (median ± MAD)**	90 ± 15.8	88 ± 4.1	0.0003
**Total cholesterol (mg/dL) (median ± MAD)**	182 ± 23.7	192.5 ± 23.1	0.0004
**HDL (mg/dL) (median ± MAD)**	58 ± 19.7	64 ± 8.1	0.0002
**LDL (mg/dL) (median ± MAD)**	113 ± 34.1	106 ± 9.9	0.0001
**Triglycerides (mg/dL) (mean ± SD)**	90 ± 141.2	77.5 ± 17.3	0.0002

## Data Availability

The raw data presented in this study are available on request from the corresponding author. The data are not publicly available due to privacy (sensible data of both patients and controls with names and surnames).

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
