# Peer review of "Metabolic Comorbidities in Vitiligo: A Brief Review and Report of New Data from a Single-Center Experience"

_ijms, 2021, doi:10.3390/ijms22168820_

Round 1
Reviewer 1 Report
Proposed paper is interesting and well written. However, some revisions are needed before it can be accepted for pubblication:
- Since a part of new data presentation is included at the end of the review it need to be stated also in the title; something like "...: a brief review and new data from our centre".
- Are there any data on atheroslerosis and target organ damage in vitiligo? I failed to found it, if not it should be proposed as a new area for future studies. In fact, as already mentioned in the following paper (High Blood Press Cardiovasc Prev. 2019 Jun;26(3):175-182. ), inflammation is strongly related to atherosclerosis.
Author Response
Thank you for the consideration of our work
We have changed the title as suggested.
We have underlined in the text that atherosclerosis could be a new area for future studies citing the suggested paper.
Overall, english language has been checked.
Reviewer 2 Report
A really timely and important review. We are only about to start associate some facts in the interaction between pigmented cells and the organism. I would like to note two problems which may or may not be mentioned in this review as concerning METABOLIC comorbidities in vitiligo. Namely the "brain-immune system-pigmented cells" axis. Perhaps there can be found a place in this review to include these interesting comorbidities?
- The authors mention several times cross-talk between vitiligo and psoriasis, the latter being an intriguing, not-only-dermatologic pathology. And what about alopecia areata, another clearly autoaggressive pathology? See e.g. Rork et al doi: 10.1097/MOP.0000000000000375 and other papers by this team, including intriguing data on the relation between vitiligo and AA in the context of MHC-I presentation in skin and in hair follicles.
- A very interesting research on the Vogt-Koyanagi-Harada syndrome and relation of vitiligo and aseptic meningitis, see e.g. Goldgeier et al., 1984 doi: 10.1111/1523-1747.ep12260111 and further.
A minor English error found when reading the text: line 190 "seem not to be limited"
Author Response
Thank you for your comments on our manuscript, we appreciate.
We have added for a brief section on the association between alopecia areata's pathogenesis, vitiligo, and metabolic co-morbidities Regarding the "brain-immune system-pigmented cells" axis it is certainly an important topic to be evaluated among the vitiligo co-morbidities.. Regarding VKH disease, we believe that this arguement would not be in the aim of the review as we would like to specifically focus on the metabolic abnormalities in vitiligo.